# A Reinforcement Learning Approach to Effective Forecasting of Pediatric Hypoglycemia in Diabetes I Patients: an extended de Bruijn Graph

## Abstract

Pediatric diabetes I is an endemic and an especially difficult disease; indeed, at this point, there does not exist a cure, but only careful management that relies on anticipating hypoglycemia. The changing physiology of children producing unique blood glucose signatures, coupled with inconsistent activities, *e.g.*, playing, eating, napping, makes "forecasting" elusive. While work has been done for adult diabetes I, this does not successfully translate for children. In the work presented here, we adopt a reinforcement approach by leveraging the de Bruijn graph that has had success in detecting patterns in sequences of symbols–most notably, genomics and proteomics. We translate a continuous signal of blood glucose levels into an alphabet that then can be used to build a de Bruijn, with some extensions, to determine blood glucose states. The graph allows us to "tune" its efficacy by computationally ignoring edges that provide either no information or are not related to entering a hypoglycemic episode. We can then use paths in the graph to anticipate hypoglycemia in advance of about 30 minutes sufficient for a clinical setting and additionally find actionable rules that accurate and effective.

## 1 Introduction

Fear of hypoglycemia and hypoglycemia by itself are major psychological and physiological barriers to achieving optimal glycemic outcomes. Self-Monitoring of hypoglycemia is a key component of diabetes care, along with carbohydrate counting and adjusting insulin dose. Hypoglycemic events include all episodes of a plasma glucose concentration low enough to cause symptoms and/or signs, such as dizziness, fast heartbeat, shaking, sweating, hunger, *etc*. Traditionally, glucose value $<$ 3.9 mmol/L (70 mg/dl) is used as the clinical alert or threshold value for initiating treatment for hypoglycemia because of the potential for glucose to fall further and avoid consequences of glucose levels below 3 mmol/L (54 mg/dl). Children with type 1 diabetes (T1D) should spend less than 4% of their time $<$ 3.9 mmol/L (70 mg/dl) and less than 1% of their time $<$ 3 mmol/L (54 mg/dl). Overtreating hypoglycemia even before reaching low is very common, and it is associated with fear of hypoglycemia. Several studies have analyzed that there is a need to lower the threshold for hypoglycemia (60 or 65mg/dl), in order to stabilize the glycemic variations. In addition, new stable insulin formulations (rapid and shorter acting insulin as well stable and flatter long-acting insulin) contribute to less variations. Glucose sensors and automated insulin delivery systems may also benefit in reducing hypoglycemic events due to different algorithms to predict hypoglycemia and adjust insulin dose. Thus, a critical use of artificial intelligence (AI) would be to improve *clinical* forecasting of blood glucose (BG) levels in pediatric T1D, since children are particularly vulnerable to a disease endemic to the entire world.

The contributions in this work are: developing a novel, effective reinforcement learning approach to address the critical problem of pediatric T1D forecasting well-enough that a solution could exist in real clinical settings. We begin with the de Bruijn graph (dBG) (extending the definition of Markov Decision Processes) by creating an alphabet over patient temporal BG sensor readings and integrate an input window reflecting numerical intervals, rather than using a single state context to make a prediction as has been traditionally done. We then tune, what we call the resolution,

graph by pruning out the less relevant parts of the dBG. Then, as sensor data is given, the graph can forecast when a hypoglycemic episode will subsequently occur within 30 minutes with acceptable clinical accuracy. Our approach demonstrates both the model effectiveness and run-time efficiency of building essentially a kind of state machine that can recognize subtle changes and, when run as a generator, can produce actionable patterns. Further, since this is a lazy structure, a new patient with limited data find the best set of graphs and then modify them to suit her needs. Fig. 1 (II.-IV.) highlights the process using a small data sample.

This paper is structured as follows: Section 2 provides the background. Section 3 details our dBG. Experimental results and conclusion are presented in Sections 4 and 5, respectively.

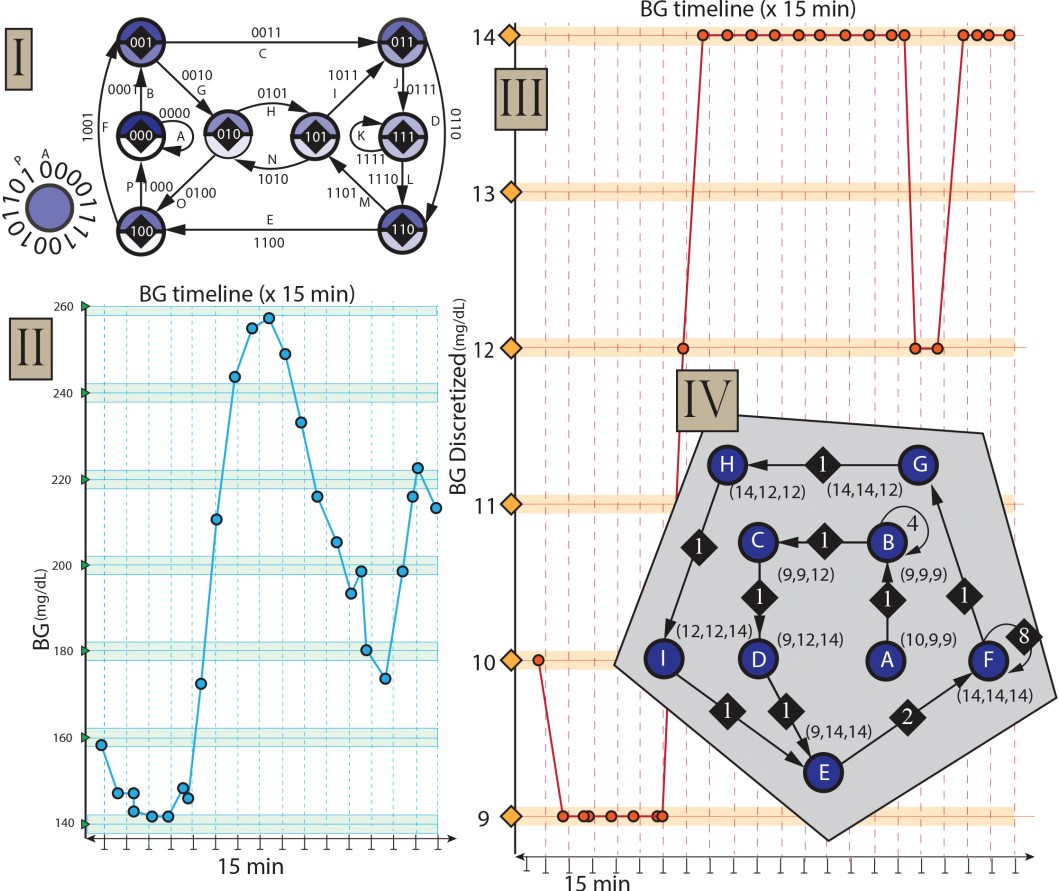

Figure 1: I. A dBG for 4-tuples over $\Sigma = \{0, 1\}$. Observe the Eularian path (A,B,. . ., P) that captures all tuples. Many properties exist like reflection of graph. The de Bruijn sequence itself is lower left, A the start and P the end and is the most efficient encoding. By moving of this so-called drum, each 4-tuple can be recovered. II. A snapshot of 15 minute intervals of a pediatric patient BG levels. We selected one that was reasonably easy to visualize, but even this one exhibits challenges in data. III. A representation of the discretization and symbolics for II. Observe that the sharp changes in II. are still captured representing idiosyncrasies of the patient–perhaps exercise, eating, napping, signal problems. IV. This shows a portion of our dBG representation. The path is shown (A,B,. . .,I) given unique four-tuples with edges showing counts that characterize patterns beginning with 10.

## 2 BACKGROUND AND RELATED WORK

Predicting both short and long-term BG concentration is crucial in diabetes management. Access to reliable algorithms for forecasting BG and hypoglycemic episodes enables proactive treatment and refined carbohydrate control, potentially preventing critical events. Despite the advancements

in classification- or regression-based methods (Faccioli et al., 2022; Yang et al., 2022; Prendin et al., 2021), a universally effective prediction approach remains elusive, furthering the need for exploration and innovation. Current research predominantly employs Continuous Glucose Monitoring (CGM) data along with other physiological parameters for prediction, but a limited number focus exclusively on CGM data. This gap is even more pronounced in pediatric research, where managing hypoglycemia is compounded by the unpredictability of children's routines and hormonal changes leading to insulin resistance (Coolen et al., 2021). Earlier studies present diverse methods for BG prediction, *e.g.*, autoregressive moving-average models and various machine learning algorithms (Eren-Oruklu et al., 2010; Paul & Samanta, 2015; Alfian et al., 2020b; Duckworth et al., 0). While recent models employing neural networks and ensemble machine learning show promise (Alfian et al., 2020a; Aliberti et al., 2019; Syafrudin et al., 2022; Mhaskar et al., 2017; Piersanti et al., 2023), the complexity of pediatric diabetes is proving harder to model; some research aims at improvement by constraining time periods, *e.g.*, nocturnal hypoglycemia prediction models and variable importance plot feature selection (Dave et al., 2021; Tkachenko et al., 2016). The most significant work we found studied ten *virtual* pediatric patients (D'Antoni et al., 2022) using recurrent Neural Networks (RNNs) and Long Short-Term Memory (LSTM) networks. While the results were promising, the nature of the data makes clinical evaluation difficult at best and adoption unlikely. Reinforcement learning (RL) (van Otterlo & Wiering, 2012) is now the most popular AI in ubiquitous use due to its generality across divers problems. RL is based, from one powerful perspective, on Markov decision processes that characterize a problem into searching through states sequentially with limited feedback.

We chose to use the dBG (van Lint & Wilson, 1992) originally a structure used to efficiently encode sequences as a Eularian path. dBG have been adopted as approaches across a number of domains where the problem can be cast a sequence of events, cryptology (Fredricksen, 1982; Lempel, 1979), sequence complexity (Games & Chan, 1983; A.H. Chan, 1982; Etzion & Lempel, 1984), networks (Samatham & Pradhan, 1989), genomics (especially assembly) (M.J. Chaisson & Pevzner, 2008; P.E.C. Compeau & Tesler, 2011; Idury & Waterman, 1995; Li & Waterman, 2003; Mahadik et al., 2019; Zhang & Waterman, 2003; Samatham & Pradhan, 1989). In particular, after construction, the dBG can characterize subtle patterns of events (or symbols) either as a recognizer or producer (much like a regular expression and its equivalent deterministic finite automaton). We imagined treating the BG levels as a sequence of events (symbols) and investigate how well a dBG can capture and detect patterns.

For an alphabet $|\Sigma| = 2$ and positive integer $k$, (de Bruijn, 1946) describes a recursive algorithm to generate the most efficient encoding of all $k$ sequences $\Sigma^k$ as a single sequence, called a de Bruijn sequence (2,$k$), $s = \sigma_1\sigma_2\cdots\sigma_\ell$ where each substring $\sigma_{i-k+1}\cdots\sigma_i$ occurs only once in $s$ modulo $\ell = 2^k$. For example, $1_0 1_1 0_2 0_3$ describes $k = 2$ and $i, i + 1 \bmod 2^k$. Fig 1 I. shows the de Bruijn graph for 4-tuples. The result was discovered both earlier (Marie, 1894) and simultaneously (Good, 1946), but de Bruijn's approach using graphs proved the most accessible.The power of this representation is that by traveling the Eularian path, every $k$-tuple (in this case 4) is encountered. The sequence is shown as the usual drum bottom left. The vertices are all possible sequences of length $k$, while the edges represent shared subsequences from the directed edge pair, the suffix and prefix, respectively. The alphabet size was soon extended to an arbitrary, but finite size. The associated graph is written $G_k(\Sigma)$ is defined with vertices $V = \{\sigma \in \Sigma^{k-i}\}$ and edges $E = \{(s_m t, t s_n) | s_m t, t s_n \in V\}$.

## 3 METHODS

### 3.1 DATA, TRANSFORMATION, AND DISCRETIZATION

This research employs data from 15 pediatric diabetes patients, totaling 22,291 BG level measurements. The hypoglycemia threshold is set at 70 mg/dL. Despite a general 15-minute sample rate, the dataset exhibits both variations and notable gaps. Gaps exceeding 20 minutes are processed as separate sequences during modeling. These gaps have numerous causes that are simply a part of any sensor of this ilk. Comprehensive patient information is tabulated in Table 1.

The dBG model necessitates initial data discretization, operating on an alphabet $\Sigma$ as opposed to BG in $\mathbb{R}_{\geq 0}$.

The model implementation demands discretization to derive an alphabet from the raw dataset. We use a uniform method, with all normoglycemia ($70 \leq v \leq 180$) alphabet characters corresponding to a 10-unit value range. This approach aims to optimize the model performance by minimizing alphabet size while maintaining dataset trends. The model assigns 12 characters for normoglycemia, two for hypoglycemia, and one for hyperglycemia, totaling 15 characters. Despite the focus on normoglycemia predictions, multiple labels within this range ensure minimal information loss post-discretization. During hypoglycemic or hyperglycemic episodes, further predictions are irrelevant, thus limiting the labels in these ranges. An additional hypoglycemia label enhances resolution in this region. The defined labels, covering broader ranges for hypo/hyperglycemia, are detailed in Table 2.

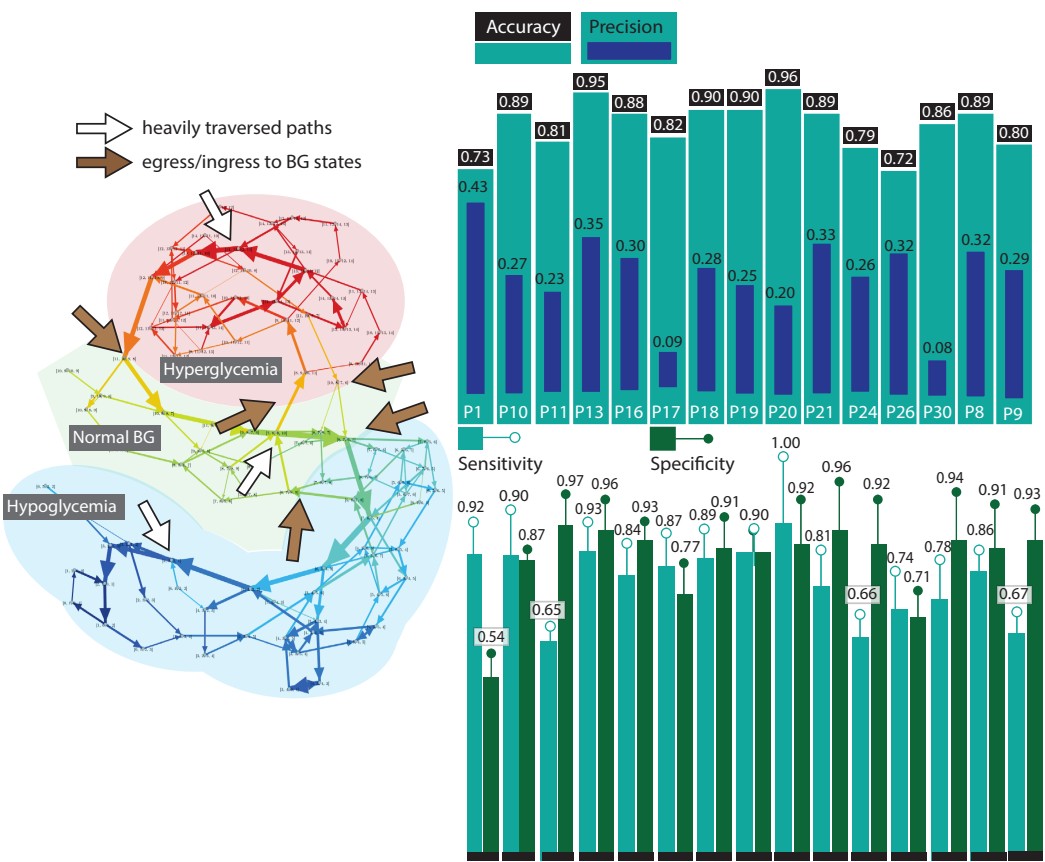

Figure 2: (LEFT) A high-level visualization of a dBG with 97 nodes and 167 edges using all available patients. The dBG is pruned with 10 uniform weight threshold. There are three BG regions: hyperglycemic, normal, and hypoglycemic. Dark blue nodes correspond to hypoglycemic values, while red nodes denote hyperglycemic values. (White arrows) Edge thickness indicates the frequency of the $k$-mers, offering visual insight into their prevalence. (Brown arrows) There are only a few paths between the three regions and interestingly, all paths must path through normal BG. These ingress/egress paths are at the kernel of rules for anticipating hypoglycemia. (RIGHT) Evaluation of patient results using leave-one-out cross-validation. We assessed the Balanced Accuracy, Precision, Sensitivity, and Specificity for all patients employing the leave-one-out cross-validation method. The average scores across all patients were 0.844, 0.268, 0.801, and 0.887 for these metrics, respectively. These evaluations were conducted using adaptive pruning and with the parameters $k = 6, \rho = 0.2, x = 6$.

| Patient | Datapoints (% Hypo.) | Study Days | Age | Gender | Weight (kg) | Height (cm) | BMI | Years With T1D |
|---|---|---|---|---|---|---|---|---|
| P1 | 1422 (63.7 %) | 18 | 16 | M | 142 | 168 | 50.3 | 3 (T2D) |
| P8 | 1325 (7.92 %) | 17 | 14 | M | 60 | 166 | 21.8 | 7 |
| P9 | 1530 (3.53 %) | 14 | 11 | F | 50 | 147 | 23.1 | 4 |
| P10 | 1444 (10.66 %) | 14 | 16 | F | 50 | 163 | 18.8 | 12 |
| P11 | 1686 (0.95 %) | 14 | 14 | F | 49 | 153 | 20.9 | 3 |
| P13 | 826 (1.94 %) | 18 | 13 | F | 40 | 146 | 18.8 | 2 |
| P16 | 1395 (5.09 %) | 14 | 13 | M | 40 | 150 | 17.8 | 12 |
| P17 | 1642 (1.28 %) | 14 | 11 | M | 50 | 151 | 21.9 | 3 |
| P18 | 1929 (6.48 %) | 19 | 15 | F | 73 | 153 | 31.2 | 7 |
| P19 | 1318 (10.09 %) | 14 | 16 | F | 56 | 155 | 23.2 | 7 |
| P20 | 719 (7.79 %) | 20 | 14 | F | 64 | 163 | 24.1 | 4 |
| P21 | 1014 (9.27 %) | 20 | 14 | F | 64 | 155 | 26.6 | 8 |
| P24 | 2631 (5.13 %) | 15 | 14 | M | 55 | 183 | 16.4 | 2 |
| P26 | 2089 (14.60 %) | 21 | 8 | F | 26 | 126 | 16.4 | 4 |
| P30 | 1321 (0.53 %) | 14 | 15 | F | 61 | 161 | 23.5 | 4 |

Table 1: Patient information for the study. T2D stands for type 2 diabetes.

| Attribute | (0-60] | (60-70] | (70-80] | (80-90] | (90-100] | (100-110] | (110-120] | (120-130] | (130-140] | (140-150] | (150-160] | (160-170] | (170-180] | (180-190] | (190-500] |
|---|---|---|---|---|---|---|---|---|---|---|---|---|---|---|---|
| Label | 0 | 1 | 2 | 3 | 4 | 5 | 6 | 7 | 8 | 9 | 10 | 11 | 12 | 13 | 14 |
| Frequency | 1418 | 872 | 878 | 1122 | 1471 | 1424 | 1265 | 1084 | 1035 | 957 | 1050 | 1005 | 944 | 826 | 6940 |

Table 2: Discretization labels and their frequency in the dataset. The 0 and 1 labels represent hypoglycemia values.

## 3.2 ALGORITHMS FOR MODEL CONSTRUCTION

The model begins by constructing a dBG from the discretized data input, denoted as $G_k(\Sigma)$, where $\Sigma$ is a given alphabet of the sequences. In the dBG, each node represents a unique $(k-1)$-mer sub-sequence from the original sequence, and each edge signifies a sequential transition between two $(k-1)$-mers, forming a $k$-mer. Edge weights indicate the $k$-mer transition frequency within the dataset. The choice of $k$ influences the dBG's structure. In our model, $(k-1)$-mers serve as the input window, utilized for future predictions. Fig. 2 (left) illustrates a high level visualization of a graph generated with all patients.

### 3.2.1 GRAPH PRUNING

The analysis emphasizes frequent transitions, necessitating dBG pruning to exclude anomalies. Despite the simplicity of flat pruning, applying a universal threshold, as visualized in Figure 2, may not be optimal. It disregards the varying significance of edges in predicting imminent hypoglycemia episodes. Hypoglycemia nodes ($V_{hypo}$) can be defined as a set of nodes that has a hypoglycemia value inside its tuple (0 or 1 in our case). In the dBG, edges that are closer to any $V_{hypo}$ node are more valuable therefore we should apply a less harsh pruning condition for these edges. To address this, adaptive pruning is proposed, considering each edge's proximity to $V_{hypo}$ nodes to ascertain its importance.

**Definition 1.** *Given a node $v \in V$, let $\delta_v$ represent the shortest unweighted path distance to the nearest $v_h \in V_{hypo}$.*

Define $R = \{(1, t_1), (t_1, t_2), \ldots, (t_n, \infty)\}$ as a list of ordered pairs, representing non-overlapping semi-open intervals $[t_k, t_{k+1})$ where $t \in \mathbb{N}$. Let $\Theta = \{\theta_1, \ldots, \theta_n\}$ with ordered $\theta$ values. Define the mapping function $f$ for every range in $R$ to a corresponding threshold in $\Theta$, $f : \mathbb{N} \to \mathbb{N}$. We thus define $f_\theta = \theta_{i,j}$ if $t_i \leq \delta_{v_{i,j}} \leq t_j$. Edges with a weight $w < f_\theta$ can be now be pruned. For larger datasets, a continuous $f_\theta$ version is also viable. This thresholding method prioritizes the retention of vital edges that boosts predictive accuracy. See Algo. 2 for the adaptive pruning implementation using $f_\theta$. For this study we used $\theta = 1$ when $1 \leq \delta_v < 2$, $\theta = 4$ when $2 \leq \delta_v < 3$ and $\theta = 10$ when $\delta_v > 3$.

Algorithm 1: GETPROBABILITY

```
1:  procedure GETPROBABILITY(G, v_start, x)
2:      function TRAVERSE(v, step, P_path)
3:          if step > x or v ∈ V_hypo then
4:              if v ∈ T then
5:                  return P_path
6:              else
7:                  return 0
8:              end if
9:          end if
10:         W ← Σ_{(u,v)∈G.Adj(v)} w_{u,v}
11:         P_total ← 0
12:         for each (v, u) ∈ G.Adj(v) do
13:             P_total ← P_total + TRA-
                VERSE(u, step + 1, P_path × w_{u,v}/W)
14:         end for
15:         return P_total
16:     end function
17:     return TRAVERSE(v_start, 0, 1)
18: end procedure
```

Algorithm 2: ADAPTIVEPRUNING

```
1:  procedure ADAPTIVEPRUNING(G, f_θ)
2:      V_included ← ∅
3:      for each (v_i, v_j) ∈ G.edges do
4:          w_{ij} ← G.edge[v_i, v_j].weight
5:          w_θ ← f_θ(G, v_i, v_j)
6:          if w_{ij} ≥ w_θ then
7:              V_included ← V_included ∪ {v_i, v_j}
8:          end if
9:      end for
10:     G' ← G[V_included]
11:     return G'
12: end procedure
```

### 3.2.2 MAKING PREDICTIONS

To predict with a dBG, we use an input subsequence (tuple) of length $k - 1$, aiming to decide on alert issuance at the $(k - 1)^{th}$ item. This tuple is analyzed within the dBG nodes to infer the most likely subsequence trajectory. As each path sequence frequency is identifiable by edge weights, it assists in estimating the probability of reaching any node from a starting node. Our focus is on predicting a hypoglycemic episode for the patient. For each node, the likelihood of reaching a $v_h \in V_{\text{hypo}}$ node is calculated, constrained by a parameter $x$, denoting the search end either by $x$ steps or upon reaching a $v_h$ node. This probability is represented as $P(\text{hypo}_x|v)$ for a node $v$. Consider nodes $v$ (current) and $v_h$ (nearest hypoglycemia node), and let $L_{v-h}$ denote the set of all paths shorter than $x$ between $v$ and $v_h$. Define $\ell_{v-h} = \{(n_1, n_2), \ldots, (n_k, n_{k+1})\}$, $(k \leq x, n_1 = v, n_{k+1} = h)$, and $L_{v-h} = \{\ell_1, \ell_2, \ldots, \ell_t\}$. Let the edge $(u, v)$ traversal probability be $P(u, v) = w_{u,v}/\sum_{(u,v)\in G.\text{Adj}(v)} w_{u,v}$, and the path probability be $P(\ell) = \prod_{(u,v)\in\ell} P(u, v)$. The overall probability is then $P(\text{hypo}_x|v) = \sum_{\ell\in L_{v-h}} P(\ell)$. Alerts for tuple $v$ are issued as follows:

$$F_{\text{alert}}(v) = \begin{cases} \text{True} & \text{if } P(\text{hypo}_x|v) > \rho, \\ \text{False} & \text{otherwise.} \end{cases} \tag{1}$$

Refer to Algo. 1 for the detailed computation of the probability for each node. With these probabilities, predictions become straightforward. We introduce a threshold parameter, $\rho$, to trigger an alert for potential hypoglycemic episodes when $P(\text{hypo}_x|v) > \rho$. $\rho$ can be adjusted to tune alert sensitivity. If the input sequence for a prediction is absent in our graph, the algorithm employs the Euclidean distance to identify and base predictions on the nearest existing node.

### 3.3 GRAPH UPDATE

The dBG model excels in its ease of update, eliminating the need for total graph reconstruction. This feature is particularly beneficial for large-scale applications. A graph database, constructed from prior patients exhibiting diverse BG traits, facilitates the seamless integration of new patient data. This new data aids in the pinpointing and enhancement of the most compatible model from the database, further fine-tuning its applicability to subsequent patients. The update process is efficient: new edges are added for non-existent tuples, and existing edge weights are incremented, all accomplished with an $\mathcal{O}(n)$ time complexity. The default weight increment is one, adjustable for more substantial graphs.

## 4 RESULTS

To evaluate our proposed model, we methodically scrutinize every datapoint in our dataset. At a given moment $t$, we assess the possible future hypoglycemic values. Let $h$ be the start of the closest hypoglycemia instance in the future. If $h - t < 1$ hour and an alert is issued at instance $t$, then it's deemed a correct prediction. Our evaluation leveraged a leave-one-out cross-validation, with each patient's data processed separately.

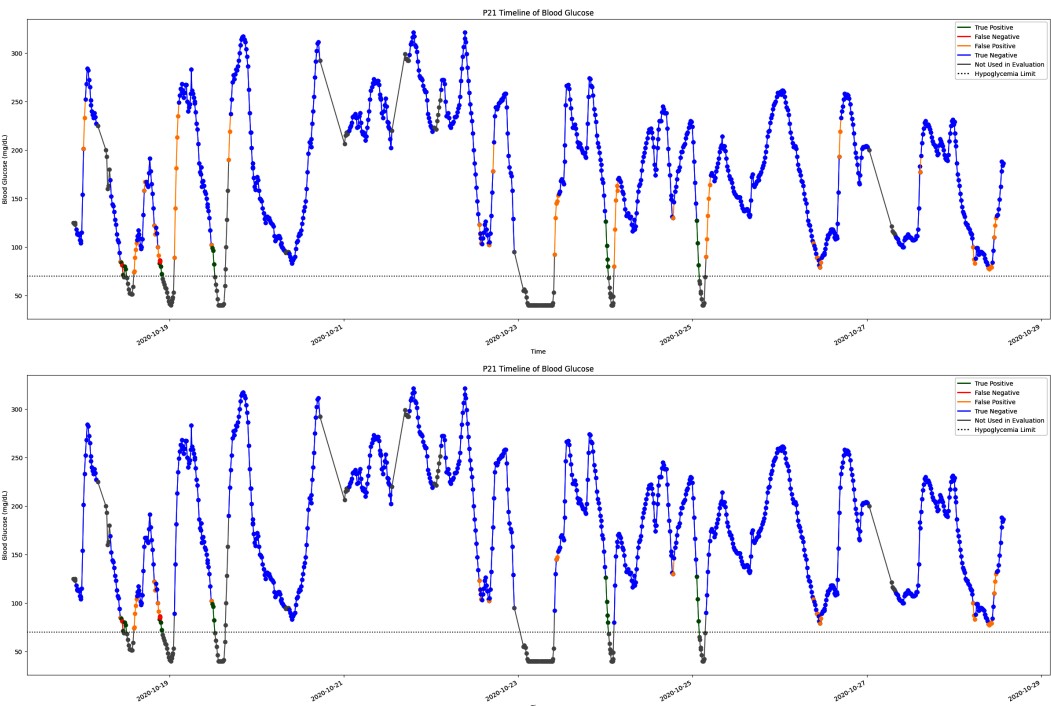

Figure 3: Timelines of alert predictions for P21. Correct and wrong predictions are marked with different colors. Gray data points are not used for evaluation, possibly due to the patient already being in a hypoglycemic state or gaps in the timeline. The upper timeline uses our dBG model, while the lower timeline uses the supervision ruleset along with the dBG model. Parameters: ($k = 6, \rho = 0.2, x = 6$).

### 4.1 MODEL SUPERVISION

Our preliminary demonstration of the model on patient P21 is depicted in the upper timeline of Figure 3. The model adeptly predicts hypoglycemic events prior to their occurrence, signified by green markers. Nonetheless, a significant limitation is the consistent generation of false alerts, denoted by yellow markers, post-return to normoglycemia from a hypoglycemic state. To mitigate these false predictions, our model is enhanced with an additional supervisory system. This auxiliary system employs a fundamental rule set $R : \mathbb{N} \rightarrow Boolean$, designed to annul the initial prediction under specific conditions. Let $S$ be the raw (un-discretized) test sequence and $i$ be the index of the datapoint that we are predicting within $S$. Then $F_{\text{alert}}^s(v, S, i) = F_{\text{alert}}(v) \wedge R(S, i)$ where $R(S, i) = (S[i] < 150 \text{ mg/dL}) \vee (S[i] - S[i-1] < 15\text{mg/dL}) \quad (i > 0)$. Observe that alerts will be suppressed if $\text{BG} > 150 \text{ mg/dL}$, or BG ascends by more than 15 mg/dL from the preceding reading.

The enhanced model, inclusive of supervisory intervention, is showcased in the lower timeline for patient P21 in Figure 3. Despite the efficacy of this rule set for the majority, it retains adaptable for tailoring to individual patient requirements for superior results. Stricter rule application may elevate the count of False Negatives while curtailing False Positives, necessitating a judicious balance to ensure an optimal trade-off.

## 4.2 Evaluation Results

The cross-validation results for each patient are detailed in Figure 2 (right). Despite the generally satisfactory performance of our model, it exhibits variability across different patients. This variation, particularly the lower scores for patients like P1, can be attributed to dataset limitations. The disproportionate hypoglycemia values in P1's data contribute to this inconsistency. However, with ample similar patient data, our model's optimization for individual patients can be enhanced. Figure 5 highlights the impact of supervision and the pruning method on our final evaluation score. Although supervision marginally diminishes sensitivity, it notably enhances specificity and precision, bolstering the alert's reliability. The pruning method markedly influences the model's performance. In total, our model preemptively issued warnings for 272 hypoglycemia instances and missed 10, achieving a 96.45% warning rate. This rate alone is not fully indicative of practical utility due to the importance of warning time. An examination of each hypoglycemia onset in our dataset reveals the majority of alerts were generated 30 minutes prior, as depicted in Figure 4. This evaluation provides a more comprehensive understanding of the model's effectiveness in real-world settings.

### 4.2.1 Limitations

A significant evaluation limitation lies in the accurate assessment of metric performance during near-hypoglycemic events. These events, where hypoglycemia is narrowly averted, often occur due to timely medication or food intake. In such scenarios, our model's alert, deemed a false positive, is in fact a valuable warning, highlighting a potential risk, as illustrated in Figure 4. The labeling of these as false positives inadvertently lowers the model's evaluated precision, despite the crucial alerts it provides for potential hypoglycemic episodes, ensuring patient safety.

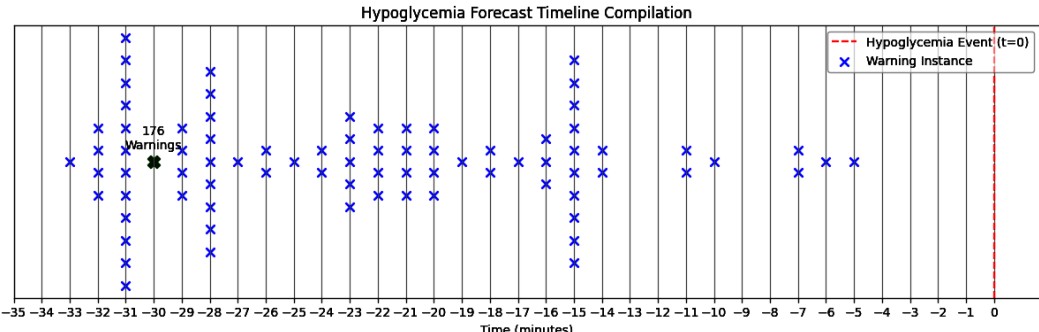

Figure 4: Forecast times for every hypoglycemia case across all patients. The figure excludes 10 instances where our model did not issue warnings for hypoglycemic cases. Despite these, our model consistently forecasts hypoglycemia with a notable 30-minute lead time in the majority of cases, as denoted by the thick marker in the figure.

## 4.3 Evaluation Across Different $k$ Values: Altering Prediction Window

We evaluated our model across various $k$ parameters, which alter the model's prediction window by modifying edge tuple length. The optimal $k$ is inherently tied to dataset size. For the current dataset, the best performance was observed at $k = 4$. Larger datasets might favor a higher $k$, due to increased likelihood of tuple overlap. Detailed results of this experimentation are provided in Table 3.

## 4.4 Graph Update Evaluation

Beyond standard leave-one-out testing, we explored graph updated evaluations to demonstrate the potential of dBGs. This experiment, although constrained by a limited dataset, showcased prospective enhancements. A single patient was chosen to represent a new patient, while the remaining dataset was divided into three groups to construct a database with three independent graphs. The selected patient's data was bifurcated; the initial half symbolized pre-treatment data, and the latter half, post-treatment data. Post-identification of the best-fit model from the database, the graph was

| k | Accuracy | Sensitivity | Specificity | Precision |
|---|---|---|---|---|
| 7 | 0.810 | 0.714 | 0.906 | 0.280 |
| 6 | 0.810 | 0.708 | 0.913 | 0.294 |
| 5 | 0.817 | 0.715 | 0.920 | 0.315 |
| 4 | 0.824 | 0.732 | 0.916 | 0.308 |
| 3 | 0.819 | 0.745 | 0.894 | 0.267 |
| 2 | 0.781 | 0.656 | 0.905 | 0.264 |

Table 3: Model performance using different $k$ parameters. No pruning is performed on the graph. $k = 2$ is structurally equivalent to a Markov chain.

updated with the evaluation data. A subsequent evaluation on the second half of the dataset was conducted. Figure 6 enumerates the outcomes with and without partial updates. Although most patients exhibited minor improvements and some major, a decline was noted in certain evaluation scores. This inconsistency is postulated to be a byproduct of dataset size limitations, with expectations of enhanced results and benefits with an extended patient sample.

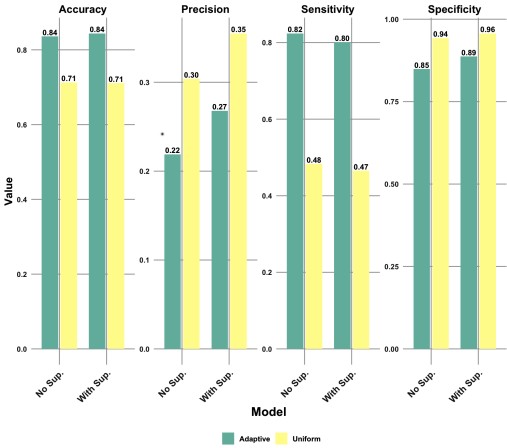

Figure 5: Model evaluation conducted with and without supervision. Weights less than 4 were pruned for uniform pruning.

Figure 6: Balanced accuracy results, and percentage accuracy differences with and without applying graph updates.

## 5 SUMMARY AND FUTURE WORK

In conclusion, our study highlights the significant potential of our extended de Bruijn graph (dBG) in accurately predicting complex systems, especially in medical settings. Our model notably outperformed in predicting hypoglycemia, showcasing its distinct efficiency. Unlike traditional machine learning techniques, dBG's low update cost makes it particularly suitable for dynamic environments like hospitals, enhancing patient care by enabling timely and precise updates. Expanding the dataset to include a more diverse patient group and extended blood glucose monitoring will enhance the dBG and its clinical applicability. With a broader and more varied dataset, the deployment of dBG across diverse clinical settings will become increasingly viable.

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
