# OpenReview forum: "A Reinforcement Learning Approach to Effective Forecasting of Pediatric Hypoglycemia in Diabetes I Patients: an extended de Bruijn Graph"
_ICLR.cc/2024/Conference — ICLR 2024 Conference Withdrawn Submission_

### Official Review · Reviewer_LJA3 · 2023-10-27

**Soundness:** 2 fair
**Presentation:** 3 good
**Contribution:** 2 fair
**Rating:** 3
**Confidence:** 5

**Summary:**

Pediatric Diabetes Type 1 is a challenging disease with no cure, requiring careful management to predict hypoglycemia due to the unique blood glucose patterns in children and their inconsistent activities. Existing approaches for adult diabetes do not translate well to children. The paragraph introduces a new method using a de Bruijn graph to transform blood glucose data into an alphabet, allowing better prediction of hypoglycemia about 30 minutes in advance, offering actionable insights for clinical use.

**Strengths:**

The authors propose a new algorithm for pediatric T1D forecasting.

The study is conducted by using real world dataset in a clinical setting.

Experimentation results are displayed clearly and soundly.

**Weaknesses:**

The topic does not serve the best for the reader of ICLR. I appreciate the authors conduct research for pediatric T1D forecasting in clinical setting, which can have significant real world impact to help patients to improve their health state. However, this focus of this study doesn't provide a more general representation method to conquer existing challenges in ML for human systems. I strongly recommend the author to submit this paper to CHIL.

Again, the methodology is not sufficiently demonstrated.

The details about the sensor data is not fully disclosed.

The challenges in methodology for pediatric T1D forecasting are weakly illustrated, making the contribution of this paper unsound.

**Questions:**

Why use RL for T1D forecasting, instead of other DL methods?

What are the hyperparameter settings? And how does the hyperparameter influence model performance?

Is the RL method robust for other dataset?

---

### Official Review · Reviewer_SBtP · 2023-10-31

**Soundness:** 1 poor
**Presentation:** 2 fair
**Contribution:** 2 fair
**Rating:** 3
**Confidence:** 3

**Summary:**

The manuscript addresses the critical problem of pediatric Type 1 Diabetes (T1D) forecasting to improve patient care. It proposes a novel reinforcement learning approach using de Bruijn graphs to predict hypoglycemic events in real-time. The work acknowledges the limitations of existing methods and aims to create a model that can adapt to individual patient needs.

**Strengths:**

The use of de Bruijn graphs and reinforcement learning for hypoglycemia prediction is innovative and distinct from traditional machine learning methods, potentially offering better results. The model can be updated efficiently, eliminating the need for total reconstruction. This adaptability is crucial in a dynamic medical environment, allowing timely adjustments to patient needs. The model demonstrates effectiveness in predicting hypoglycemic events, with a focus on alerting patients 30 minutes before an episode occurs. High prediction rates indicate its potential for improving patient safety. The discussion of rule sets for supervision and individual patient requirements shows an understanding of the need for personalized care in healthcare.

**Weaknesses:**

The manuscript mentions dataset limitations, which could affect the generalizability of the model's performance. The reliance on a single patient's data and a relatively small dataset may not fully capture the complexity of T1D and, the model still generates false alerts, particularly after a patient returns to normoglycemia from a hypoglycemic state. This may impact patient trust in the system.
Accurately assessing metric performance during near-hypoglycemic events is a limitation, and the labeling of crucial alerts as false positives can reduce the model's evaluated precision, and the choice of the optimal "k" parameter is inherently tied to dataset size, which raises questions about the model's scalability and generalizability to larger datasets.

**Questions:**

1) Given the limited dataset used in this study, how confident can we be in the model's effectiveness and reliability in real-world clinical settings? How can you address concerns about potential overfitting to the dataset?
2) The manuscript acknowledges that some patients exhibited a decline in evaluation scores with partial updates. Can this inconsistency be attributed to the limitations of the dataset, and if so, how can these limitations be addressed to ensure more consistent performance?
3) The manuscript highlights the importance of individual patient requirements. How do you plan to tailor the model to specific patient needs in a way that balances sensitivity and specificity effectively?
4) Can the authors provide more information about the scalability of the model? How well does it handle larger datasets and more complex patient scenarios, and what challenges might arise with increased data volume and diversity?
5) How does the model address the challenges of patient non-compliance or variability in their routines, which can greatly impact diabetes management and the accuracy of the predictions?

---

### Official Review · Reviewer_SYj4 · 2023-11-05

**Soundness:** 2 fair
**Presentation:** 3 good
**Contribution:** 2 fair
**Rating:** 3
**Confidence:** 4

**Summary:**

The authors introduced a reinforcement learning framework aimed at predicting episodes of pediatric hypoglycemia in Type I diabetes patients. The approach uses a de Bruijn graph to capture the various glucose level statuses of patients and assess the transition probabilities among these states. The method was tested on a dataset comprising 30 patients.

**Strengths:**

The research task explored in this paper may have high clinical significance.

**Weaknesses:**

1. The major issue is with the experimental evaluations. The authors have provided metrics such as accuracy, precision, and specificity, but there is an absence of comparisons with baseline models. This makes it challenging to determine the performance of the proposed method over conventional time-series methodologies. The author claims that 'the nature of the data makes clinical evaluation difficult at best and adoption unlikely' for RNN-based method but there is no justifications for this claim.

2. Why do the authors not use AUROC or AUPRC as classification metrics?

3. Another concern is about scalability. It is uncertain whether the proposed method can handle time series with many different changing patterns without the graph size increasing exponentially.

4. The applicability of this work to the ICLR conference is questioned. While the findings may hold clinical significance, the implications from a computer science standpoint seem limited based on the current justifications provided in the paper. The research might be more suitably presented in a clinical journal, where the focus is predominantly on medical and health-related outcomes.

**Questions:**

Please address the weaknesses above.